# The Contribution of the Hunger Hormone Leptin in the Aetiology of Postoperative Anorexia after Laparoscopic and Open Gastrectomy in Gastric Cancer Patients

**DOI:** 10.3390/biom11111601

**Published:** 2021-10-29

**Authors:** Tomaz Jagric

**Affiliations:** Department for Abdominal and General Surgery, University Clinical Centre Maribor, Ljubljanska ulica 5, 2000 Maribor, Slovenia; tomaz.jagric@gmail.com

**Keywords:** serum leptin, nutritional status, laparoscopic gastrectomy

## Abstract

Background: Laparoscopic surgery produces lesser postoperative inflammation with a smaller cytokine and leptin response, and might thus reduce postoperative anorexia compared with open surgery. The aim of the present study was to determine the role of serum leptin in postoperative anorexia after laparoscopic gastric cancer surgery. Methods: Fifty-four consecutive patients with adenocarcinoma of the stomach were operated on either with open or laparoscopic surgery. Correlations were determined between the serum levels of leptin, clinico-pathological characteristics, serum haemoglobin, and albumin. Results: Serum leptin levels on day seven were correlated significantly to gender (*p* = 0.004), body mass index (BMI) (*p* = 0.002), and tumour grade (*p* = 0.033). In the patients with C-reactive protein (CRP) < 100 mg/L (*n* = 46) the leptin levels on day seven were significantly lower after the laparoscopic operation (*p* = 0.042) and in patients with lower BMI (*p* = 0.001). The linear regression model determined a significant correlation between the relative concentration of leptin on day seven and laparoscopic surgery (Beta−0.688; *p* < 0.0001), gender, BMI, location of the tumour, T stage, N stage, perioperative therapy, tumour grade, perineural invasion, Lauren histological type, and ulceration. In patients with CRP levels below 100 mg/mL, the serum level of albumin on day seven after surgery was significantly higher in patients after laparoscopic surgery. Conclusion: Laparoscopic surgery produced significantly lower relative leptin concentrations on day seven, and higher serum albumin levels in the subgroup with CRP levels below 100 mg/L at discharge. These results suggested that laparoscopic gastric cancer surgery might reduce postoperative leptin response, leading to a better nutritional status at discharge compared with open surgery.

## 1. Introduction

Gastric cancer is the second leading cause of cancer related deaths in the world [1,2,3,4]. Surgery offers the only chance of a cure. Meanwhile, gastric cancer surgery has a significant impact on patients’ quality of life due to the extensive surgery [1,2,3,4]. Minimally invasive approaches have been introduced to reduce the tissue trauma, to expedite postoperative patients’ recovery, and improve quality of life [1,2,3,4]. According to randomized controlled trial (RCT) from the East, patients after laparoscopic gastric cancer surgery recovered faster than patients after open surgery [1,2,3]. Prospective randomised controlled trials in Japan, Korea, and China have confirmed that laparoscopy reduced the postoperative pain, reduced the time to the first bowel movement, and shortened the duration of the hospitalisation [1,2,3]. One of the main factors that determines the postoperative patient recovery is the duration of postoperative bowel dysfunction and anorexia. Anorexia is often prominent in gastric cancer patients [5,6,7]. Especially prone are patients after major resections and after perioperative chemotherapy [5,6,7]. Anorexia remains a major clinical problem, because it contributes to worse long-term survival [5,6,7] and compromises quality of life [8]. There is mounting evidence that the inflammatory response is a key factor that mediates gastrointestinal alterations, causing anorexia in cancer patients [8,9,10,11]. Among other inflammatory cytokines, leptin, an adipokine, has been identified as a potential factor in the development of anorexia [8,9,10,11]. It acts centrally in the arcuate nucleus of the hypothalamus, and inhibits the release of neuropeptide Y that, in turn, stimulates the food intake, decreases thermogenesis, and increases plasma insulin and cortisone levels [5,6,7]. As laparoscopy has been shown to reduce the tissue trauma and inflammatory response effectively after surgery [1,2,3], this might also reduce the postoperative inflammatory response, leptin secretion, and anorexia. We have, therefore, proposed that laparoscopic surgery produces lesser postoperative anorexia compared to open surgery by means of lowering the postoperative serum leptin levels. The aim of the present study was to compare the postoperative leptin levels after laparoscopic and open surgery, to compare the nutritional status at discharge, and determine the correlation between leptin and pathologic and prognostic factors that might be associated with tumour progression and prognosis.

## 2. Methods

### 2.1. Patients

Fifty-four consecutive patients operated on for adenocarcinoma of the stomach with curative intent between January 2019 and June 2021 were included in this study. All participants gave formal consented before being enrolled in the study. Before the operation, every patient had an upper gastro-intestinal endoscopy, a complete radiological work-up consisting of Computer Tomography of the thorax and abdomen, standard laboratory tests, and a chest X-ray. Demographic, antropometric, and clinical data were obtained from the participants. Patients were discussed on a tumour board. Patients eligible for perioperative treatment received 4 cycles of FLOT (docetaxel, oxaliplatin, fluorouracil, and leucovorin), consisting of docetaxel (60 mg/m^2^), oxaliplatin (85 mg/m^2^), leucovorin (200 mg/m^2^) and 5-fluorouracil (2600 mg/m^2^ as a 24 h infusion), all given on day 1. This procedure was performed every 2 weeks, and surgery followed 3 weeks after chemotherapy was completed. Patients received 4 cycles of FLOT postoperatively. Patients were operated on laparoscopically or open based on the preference of the operating surgeon. The contraindications for laparoscopy were previous upper abdominal surgery, excluding laparoscopic cholecystectomy, a BMI of more than 30 kg/m^2^, glaucoma, increased intracranial pressure, severe Chronic obstructive pulmonary disease (COPD), or asthma.

Patients were divided into two groups. The open surgery group (OS) consisted of 32 patients, and the laparoscopic surgery group consisted of 22 patients. Patients with grade IIIb morbidity or perioperative complications, mechanical bowel obstruction, or any obvious functional obstruction to food intake, or patients with acute phase response defined as having a serum c-reactive protein (CRP) concentration of more than 100 mg/L were excluded from the study. The study protocol was reviewed and approved by the Institutional Ethics Committee of the University Clinical Centre Maribor (UKC-MB-KME-45/20, approved on 7 July 2020).

### 2.2. Operations

All surgery was performed according to the Japanese Gastric cancer guidelines (JGCG) [12]. Depending on the preoperative T stage, location of the tumour, the differentiation, and the Lauren’s classification, patients received either a subtotal or a total gastrectomy. A D2 lymph node dissection was performed in all cases except in early gastric cancer.

In the laparoscopic group, patients were operated on by a single team experienced in laparoscopic gastric cancer surgery. The laparoscopic procedure was described elsewhere [13,14]. Briefly, the operations were performed in a highly standardised manner by the five port approach. The patient was in the decubitus position with the legs extended, with the surgeon and the first assistant on either side of the patient, and the second assistant between the legs. The intestinal continuity was performed with an Omega loop or with Roux-en-Y reconstruction. The specimen was extracted trough a 5 cm to 8 cm mini-laparotomy. All laparoscopic procedures were performed in a totally laparoscopic fashion.

### 2.3. Serum Leptin Detection

Serum leptin levels were determined from blood samples taken before the operation and on the seventh postoperative day. A Human Leptin Quantikine ELISA Kit (R&D Systems, Bio-Techne Corporation, Minneapolis, MN, USA) was used for serum leptin determination. The procedure was performed as specified by the manufacturer. First, 100 µL of Assay Diluent were added to each well. Next, 50 µL of sample were added to each well, covered with a plate sealer, and incubated at room temperature for 2 h on a horizontal orbital microplate shaker. Each well was aspirated and washed. The process was repeated 3 times for a total of 4 washes. Then, 200 µL of Conjugate was added to each well, covered with a new plate sealer, and incubated at room temperature for 2 h on the shaker, aspirated, and washed 4 times. 200 µL of Substrate Solution was added to each well. Plasma samples were incubated at room temperature for 30 min on the benchtop. Finally, 50 µL of Stop Solution was added to each well and read at 450 nm within 30 min.

### 2.4. Study End-Points

The primary objective was to determine the differences in postoperative serum leptin levels on day seven after the operations. Secondary end-points were the postoperative haemoglobin levels, albumin levels, CRP levels and hospital stay length. The tertiary endpoint was the correlation between the serum leptin levels and clinicopathological characteristics.

### 2.5. Statistical Analysis

Data were presented as mean + SD, median (interquartile range) or absolute number (percentage) where appropriate. The association between leptin amd the clinicopathological characteristics was determined with the Sutdent’s t-test, Mann-Whitney’s U test and the χ^2^ test. The distribution of data was determined with Kolmogorov-Smirnov’s and Shapiro-Wilk’s tests of normality. Correlations between variables were determined with the Pearson’s bivariate correlation test. A linear regression model was used to determine the variables related with the serum leptin levels. Statistical significance was defined as a *p* < 0.05. The statistical analysis was performed with the SPSS 25.0 for Windows (IBM, Armonk, NY, USA).

## 3. Results

### 3.1. Clinicopathological Comparison between LS and OS

The clinicopathological characteristics of the included patients are presented in Table 1. The clinicopathological characteristics of patients in groups laparoscopic surgery (LS) and OS are also presented in Table 1. There were no significant differences in the distribution of age, sex, BMI, location, type of resection, TNMN (tumour, node, metastasis) distribution, total number of resected lymph nodes, total number of positive lymph nodes, tumour grade, perineural invasion, lymphovascular invasion, ulceration or Lauren histological type between the groups. Patients in the OS group had a longer mean hospital stay (10 (4.5) days) compared with the LS group (8 (3.5) days), but this difference did not reach the level of significance. Patients in the OS had more T4a and T4b tumours compared with LS (*p* = 0.034). There was also a significant difference in the perineural invasion distribution (*p* = 0.019). Most of the patients in the LS group had no perineural invasion present (75%) compared with 43.3% of patients in the OS group.

In the subgroup analysis of patients (*n* = 46) with CRP levels below 100 mg/L, no significant differences were noted in the distribution of demographic or pathological characteristics. A significant difference was noted in the distribution of perineural invasion (*p* = 0.036). Patients in the LS group were without perineural invasion in 73.7% cases compared with 42.7% cases in the OS group.

### 3.2. Postoperative Serum Levels of CRP, Haemoglobin, Albumin and Leptin in LS in OS

The serum levels of CRP rose slightly from day one after surgery to day 7. There were no significant differences between LS and OS. In both groups, the levels of haemoglobin fell on day four after surgery, and then rose gradually until day seven, but were still below the base-line preoperative levels. The serum levels of albumin declined on day four compared with the preoperative levels in both groups. There were no significant differences between the groups in albumin levels on day four. From day four, the levels of serum albumin began to increase to day seven, but were still below the preoperative base-line levels. Although the levels of serum albumin were higher in the LS group compared with OS, this difference did not reach the level of significance (Figure 1).

In both groups, the serum level of leptin declined significantly compared with the preoperative base-line. The serum level of leptin fell to 68% compared with the preoperative base-line levels. Although patients in the LS group had lower levels of leptin (52 ± 59 pg/mL in LS vs. 65 ± 63 pg/mL in OS) and relative serum levels of leptin on day seven (63 ± 39.5% in LS vs. 72.6 ± 34% in OS), they did not reach the level of significance.

Similar serum dynamics of serum haemoglobin, albumin, and leptins were observed in the subgroup of patients with CRP levels below 100 mg/L. There were no significant differences in the postoperative CRP levels on day one and four between LS and OS. Patients in the LS group had significantly higher levels of haemoglobin on day four compared with OS (111 ± 13 g/L in LS vs. 100 ± 8 g/L in OS; *p* = 0.019). Patients in the LS group had significantly higher levels of albumin on day seven (30.5 ± 1.7 g/L in LS vs. 25 ± 4 g/L in OS; *p* = 0.039). Patients in the LS group also had significantly lower serum leptin levels on day seven compared with OS (30 ± 22 pg/mL in LS vs. 70.4 ± 68 in OS; *p* = 0.038). The levels of leptin fell to 50% in LS compared with OS, where the levels fell to 74% of the preoperative base-line levels (Figure 2).

### 3.3. Correlations between Leptin and Clinicopathological Predictors

Preoperative serum leptin levels were correlated significantly with patients’ BMI (*p* < 0.0001) and the presence of distant metastases (*p* < 0.0001). Similarly, in the subgroup of patients with CRP levels below 100 mg/L on day one, only BMI and the presence of distant metastases were correlated significantly with the preoperative leptin levels.

Serum leptin levels on day seven were correlated significantly to sex (*p* = 0.004), distant metastases (*p* = 0.019), patients’ BMI (*p* = 0.002), tumour grade (*p* = 0.033) and serum CRP levels on day one (*p* = 0.027). The scatter plot analysis confirmed a significant relation of serum leptin levels on day seven with CRP levels on day one (R^2^ 0.104) (Figure 2 scatter plot). Female patients were found to have significantly higher leptin levels on day seven compared with male patients (37.5 ± 32 pg/mL in males vs. 84.6 ± 76 pg/mL in females; *p* = 0.004. The scatter plot analysis showed a significant positive correlation between BMI and leptin levels on day seven (Figure 3). The correlations matrix is presented in Table 2.

In the subgroup of patients with CRP < 100 mg/L on day one, the leptin levels on day seven were correlated significantly with the laparoscopic operation (*p* = 0.042) and BMI (*p* = 0.001). R^2^ = 0.244.

Since BMI contributed significantly to the baseline values of leptin, the absolute values of leptin are difficult to compare between patients with varying BMI values. To adjust for the confounding effect of BMI on baseline leptin values, we determined the relative serum concentration of leptin on day seven. It was defined as the proportion of leptin levels on day seven compared to day one ((leptin day 7/leptin day 1)*100). The rate of leptin on day seven was not correlated significantly to either BMI or gender.

### 3.4. Multivariate Analysis

A forward stepwise linear logistic regression was performed to determine the correlations between postoperative leptin levels on day seven and the clinicopathological predictors. In the final model, the serum levels of leptin on day seven were associated significantly with distant metastases (Beta 0.548; *p* < 0.0001), female sex (Beta 0.357; *p* = 0.002), BMI (Beta 2.091; *p* = 0.003), Lauren mixed histological type (Beta 0.316; *p* = 0.004), and subtotal gastrectomy (Beta 0.298; *p* = 0.006).

A significant correlation was found between leptin levels on day seven and BMI. The relative serum levels of leptin on day seven were determined to adjust for the confounding effect of BMI on the base-line leptin levels. A backwards linear regression model was used. Laparoscopic surgery was found to be associated significantly with the relative serum levels of leptin on day seven (Beta −0.688; *p* < 0.0001). Additionally, female sex (Beta 0.619; *p* < 0.0001), BMI (Beta −0.472; *p* = 0.011), distal third tumour location (Beta 0.786; *p* < 0.0001), advanced T stage (Beta −0.962; *p* < 0.0001), early N stage (Beta 0.649; *p* = 0.003), perioperative therapy (−0.859; *p* = 0.001), low/high tumour grade (Beta 0.596; *p* < 0.0001), perineural invasion (Beta 0.572; *p* = 0.001), Lauren diffuse histological type (Beta 0.691; *p* < 0.0001), absence of ulceration (Beta −0.251; *p* = 0.027), and distal resection margin (Beta 1.362; *p* < 0.0001) were associated significantly with the relative serum levels of leptin on day seven.

## 4. Discussion

The aim of the present study was to determine whether laparoscopic surgery could promote a shorter duration of postoperative anorexia compared with open surgery. As leptin is one of the key regulators of appetite, we compared the serum levels of leptin between patients operated on with laparoscopic and open surgery. We also determined the correlation between the type of operation, length of hospital stay, nutritional status of the patients at the time of discharge, and serum levels of leptin.

In the present study, 54 consecutive gastric cancer patients were operated on, either with open or laparoscopic surgery with curative intent. Both groups were comparable according their base-line demographic and pathologic characteristics. Patients in both groups had comparable preoperative serum leptin levels. In both groups, serum levels of leptin fell in the postoperative period. This was expected and in line with previous observations [15]. Gómez-Ambrosi et al. described a relation between leptin and other adipokines, specifically serum amyloid A (SSA) that stimulates leptin secretion. SSA is produced mainly in the omentum. Omentectomy was performed as a part of radical gastrectomy in our study, whereby the predominant site of SSA synthesis was removed. The consequent drop of SSA production could explain the postoperative drop in serum leptin levels. Although patients in the LS group had lower serum levels of leptin on day seven, this difference did not reach the level of significance. However, in patients with CRP levels below 100 mg/L, the difference was significant. This was an interesting finding, as it points to a possible correlation of postoperative serum leptin levels with systemic inflammatory response. Inflammatory cytokines have been suggested to influence postoperative serum levels of leptin [8]. Tuzun et al. described that leptin levels were higher in patients with active ulcerative colitis, and its serum levels were related to the extent of inflammatory colon involvement [15]. A Leptin receptor is related to class I cytokine receptors, which might explain why inflammatory response could influence the leptin’s serum concentration [9,16]. This was in line with our observations, as we determined a strong correlation between serum CRP levels on day one and serum leptin levels on day seven. Many RCT could confirm that laparoscopy causes significantly less tissue trauma and smaller systemic inflammatory response [1,2,3]. Hence, the difference in serum leptin levels between LS and OS groups could, therefore, be related to the smaller-level tissue trauma after laparoscopy, and the lesser extent of systemic inflammatory response. In the present study, we could confirm that the smaller inflammatory response after laparoscopic surgery is related to lower serum levels of leptin on day seven. On the other hand, the results also suggested that any beneficial effect of laparoscopy on postoperative nutritional status can only be maintained as long as there are no postoperative complications. In patients with complications and increased CRP levels, the advantages on the nutritional status of laparoscopy over open surgery are lost.

The higher levels of leptin might cause more pronounced and prolonged anorexia after open surgery compared with laparoscopic surgery. To investigate whether serum leptin was related to the nutritional status of the patients, we observed serum levels of haemoglobin and album after laparoscopic and open gastrectomy and its correlation with serum leptin levels. Both serum haemoglobin and albumin are surrogate markers of the nutritional state of patients [17,18]. We found no significant differences in haemoglobin levels between the groups on day seven. On the other hand, we noted higher serum albumin levels in LS patients on day seven. This difference was significant in the subgroup of patients with CRP levels below 100 g/L. This result confirmed that patients after laparoscopic surgery and CRP levels below 100 g/L had a better nutritional status at the discharge compared with open surgery. The precise mechanism by which hypoleptinemia exerts its activity on the nutritional status is complex due to the pleiotrophic effect of leptin [19]. Hypoleptinemia might directly stimulate nutrient intake; on the other hand, leptin may promote processes at least partly independent of its anorexic effect. Many reports have confirmed cross-talk between leptin, insulin, and protein pathways [19]. Shimabukuro has reported that leptin constrains insulin biosynthesis and secretion in pancreatic β-cells and lower insulin [20]. Insulin is known to stimulate muscle cell protein synthesis [20]. This might explain lower albumin levels in patients with higher postoperative leptin in the OS group.

Despite the better nutritional status of the patients in LS with CRP levels below 100 mg/L, no significant difference could be observed in the hospitalisation length. Similarly, no differences in the hospital stay were observed between open and laparoscopic gastric cancer surgery in recent Western RCT [4]. Van der Wielen et al. concluded that laparoscopy did not reduce the hospital stay compared to open surgery [4]. Our results are in line with those observations, and we agree that patients operated with both open and laparoscopic approaches can be maintained on the same postoperative recovery protocol, with only minor differences in regard to the hospital stay. However, the hospital stay does not say anything about the patients’ nutritional status, their general condition, or their quality of life at discharge. We believe that serological markers provide a better picture of the nutritional state of the patient, and have prognostic relevance and are more important than hospital stay.

The linear logistic regression model failed to identify laparoscopic surgery as significantly related to serum leptin levels on day seven. As leptin is produced mainly in adipose tissue [5,6,7], patients with higher BMI and women who have more adipose tissue produce higher baseline serum levels of leptin. Gómez-Ambrosi et al. reported that obesity was associated with a state of low-grade chronic inflammation, whereby increased secretion of different adipokines was observed [15]. These might be directly responsible for the increased circulating leptin levels in obese patients. BMI and sex carried a confounding bias in our analysis, which was confirmed by a significant correlation between serum leptin, BMI, and sex in our study. Therefore, we used the relative serum level of leptin on day seven, which provided a better measure of the serum leptin response independent of its preoperative base-line levels. The results of the linear regression model indicated that laparoscopic gastrectomy was related to lower relative serum levels on day seven compared with open gastric cancer surgery. To the best of our knowledge, this is the first study that related postoperative serum leptin levels to laparoscopic gastric cancer surgery and its role in the aetiology of postoperative anorexia.

The linear regression model also identified female sex, BMI, advanced T stage, perioperative therapy, perineural invasion, ulceration, and the distal resection margin to be significantly related to the relative serum levels of leptin on day seven. Most of these clinicopathological factors have already been identified as related to serum levels of leptin or higher leptin expression in the tumour cells [21,22,23,24]. Our results are in line with those observations.

On the other hand, the linear regression model identified a significant correlation between higher relative serum levels of leptin on day seven and the distal tumour location, early N stage, Lauren diffuse type, and higher tumour grade. These are in contradiction to results published in other studies [21,22,23,24]. Leptin was found to be overexpressed in the intestinal type of gastric cancer and in well-differentiated tumours [21,22,23,24]. Higher leptin expression was found in advanced gastric cancer and proximal tumour locations [21,22,23,24]. It should be noted, however, that, in former studies, the correlation was evaluated between preoperative serum leptin levels and clinicopathological parameters. The preoperative serum levels of leptin were not influenced by the inflammatory response after the operation. In our study, on the other hand, we were interested in the postoperative serum leptin dynamics and their relation to postoperative anorexia. The results of the linear regression model should, therefore, be viewed with caution.

Our study is not without limitations. The sample size was relatively small, and further prospective studies on a larger sample size might be necessary to confirm the relations between leptin and anorexia. In addition, monitoring of other markers that would link leptin to nutrient intake was not included in the present study. Weight recovery after operation could better determine the relation between nutrient intake and serum leptin levels. It has to be pointed out that appetite and metabolism regulation in humans is a complex and multifactorial process. It is, therefore, unlikely that it is mediated solely by leptin. Many other mediators of postoperative anorexia have been proposed [5,6,7]. Ghrelin has been extensively studied and has been found to have a prominent role in hunger modulation after bariatric operations. However, its role after gastric cancer surgery is questionable. Ghrelin is mainly produced in the gastric fundus, which is resected during total gastrectomy. Wang et al. have demonstrated a sharp decrease in serum ghrelin concentrations in the early postoperative period after gastrectomy and a slow increase in the following 360 days after surgery [25]. Therefore the role of ghrelin in early postoperative hunger and anorexia control should be negligible, which is not to say that other mediators are involved. Further studies of neuro-humoral responses and anorexia after laparoscopic surgery will be necessary to determine the exact role of leptin in the development of postoperative anorexia.

In conclusion, the present study confirmed that laparoscopic surgery was related significantly to lower relative serum levels of leptin on day seven, which, in turn, promoted significantly higher serum albumin levels at discharge in the subgroup of patients with CRP levels below 100 mg/L compared with open surgery. Laparoscopy could, therefore, promote a better nutritional status in a subgroup of patients by lowering serum leptin levels after operation, and reducing postoperative anorexia compared with open surgery. The precise mechanism of leptin’s action after laparoscopic surgery remains to be determined in future studies.

## Figures and Tables

**Figure 1 biomolecules-11-01601-f001:**
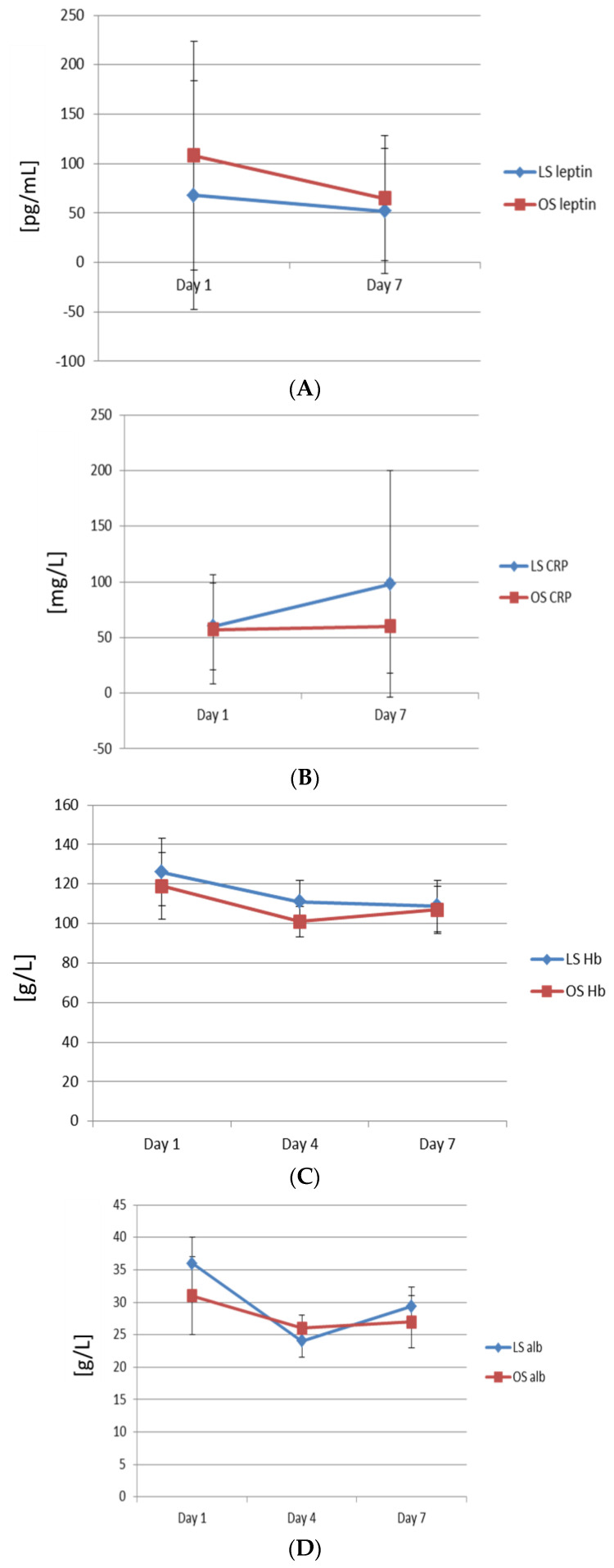
Postoperative dynamics of serum leptin, CRP, haemoglobin, and albumin between groups. (**A**) Postoperative dynamics of serum leptin concentrations. (**B**) Postoperative dynamics of serum CRP. (**C**) Postoperative dynamics of serum haemoglobin. (**D**) Postoperative dynamics of serum albumin. LS: laparoscopic surgery; OS: open surgery.

**Figure 2 biomolecules-11-01601-f002:**
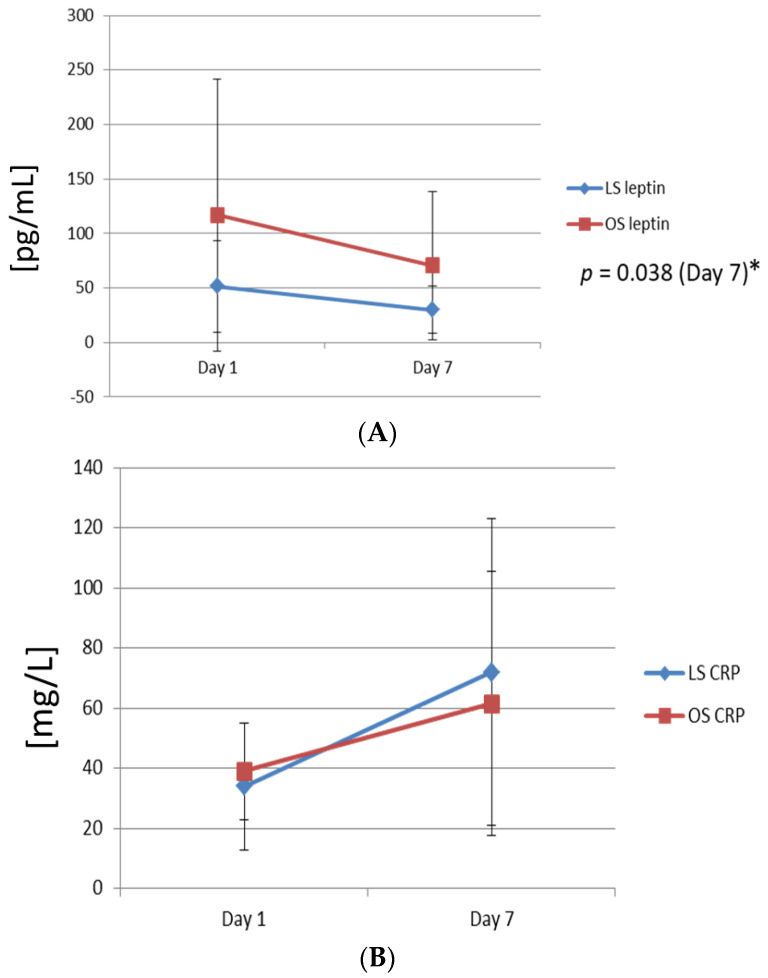
Postoperative dynamics of serum leptin, CRP, haemoglobin, and albumin between groups in the subgroup of patients with serum CRP < 100 mg/L. (**A**) Postoperative dynamics of serum leptin concentrations. * Serum levels of leptin on day seven were significantly lower in the LS group. (**B**) Postoperative dynamics of serum CRP. (**C**) Postoperative dynamics of serum haemoglobin. (**D**) Postoperative dynamics of serum albumin. LS: laparoscopic surgery; OS: open surgery.

**Figure 3 biomolecules-11-01601-f003:**
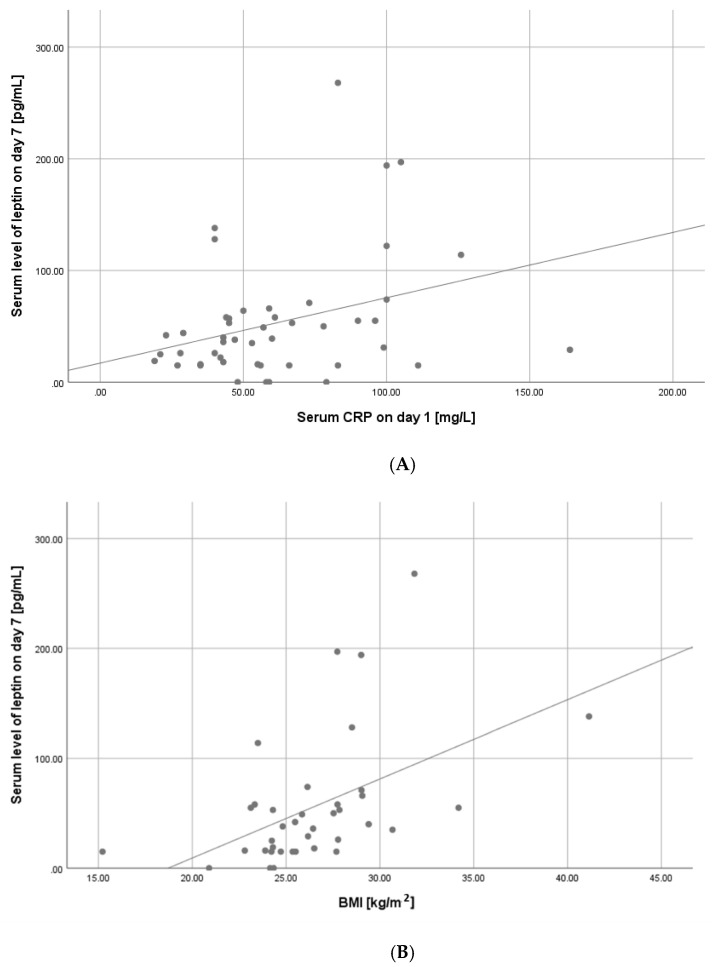
(**A**) Correlation between serum level of leptin on day seven and CRP levels on day one. R^2^ = 0.104; (**B**) Correlation between serum level of leptin on day seven and BMI.

**Table 1 biomolecules-11-01601-t001:** Clinicopathological characteristics.

Variable	All Patients (*n* = 56)	All Patients		*p*	CRP < 100 mg/mL		*p*
		OS (*n* = 32)	LS (*n* = 24)		OS (*n* = 27)	LS (*n* = 19)	
Age [years ± SD]	69 ± 14	69 ± 12	68 ± 12	NS	68 ± 13	67 ± 11	NS
Sex [*n* (%)]				NS			NS
Male	37 (66)	22 (68.8)	15 (62.5)		18 (66.7)	15 (78.9)	
Female	19 (34)	10 (31.3)	9 (37.5)		9 (33.3)	4 (21.1)	
Tumour location [*n* (%)]				NS			NS
Proximal third	12 (21.4)	10 (31.3)	2 (8.3)		8 (29.6)	2 (10.5)	
Middle third	24 (42.9)	11 (34.4)	13 (54.2)		11 (40.7)	11 (57.9)	
Lower third	20 (35.7)	11 (34.4)	9 (37.5)		8 (29.6)	6 (31.6)	
Distal resection margin [mm ± SD]	61 ± 4	55 ± 38	68 ± 41	NS	64.7 ± 42	81.5 ± 40	NS
BMI [kg/m^2^ ± SD]	25.3 ± 1	28 ± 4	25 ± 2	NS	29 ± 5	25 ± 2	NS
Hospital stay [days (IQR)]	9 (4)	10 (4.5)	8 (3.5)	NS	10 (3.5)	9 (4)	NS
Type of resection [*n* (%)]				NS			NS
Total gastrectomy	41 (74.5)	25 (80.6)	16 (66.7)		23 (88.5)	14 (73.7)	
Subtotal gastrectomy	14 (25.5)	6 (19.4)	8 (33.3)		3 (11.5)	5 (26.3)	
Leptin [pg/mL ± SD]							
Preopeative	88 ± 20	108 ± 116	68 ± 56	NS	117 ± 125	51.6 ± 42	NS
Day 7	32 ± 9	56 ± 63	52 ± 59	NS	70.4 ± 68	30 ± 22	0.038
Relative serum levels of leptin [% ± SD]	68 ± 37	72.6 ± 34	63 ± 39.5	NS	74 ± 35	50 ± 31	NS
CRP day 1 [mg/L ± SD]							
Day 1	59 ± 31	57 ± 49	60 ± 39	NS	39 ± 16	34 ± 21	NS
Day 7	65 ± 13	60 ± 42	98 ± 102	NS	61.5 ± 44	72 ± 51	NS
Haemoglobin [g/L ± SD]							
Preoperative	117 ± 8	119 ± 17	126 ± 17	NS	119 ± 18	127 ± 17	NS
Day 4	105 ± 11	101 ± 7.7	111 ± 11	0.011	100 ± 8	111 ± 13	0.019
Day 7	108 ± 12	107 ± 12	109 ± 13	NS	107 ± 12	110 ± 12	NS
Albumin [g/L ± SD]							
Preoperative	36 ± 5	31 ± 6	36 ± 4	NS	33 ± 6	39 ± 2.7	NS
Day 4	25 ± 1.5	26 ± 2	24 ± 2.5	NS	25.6 ± 2	26 ± 1.8	NS
Day 7	27 ± 3.6	27 ± 4	29.4 ± 3	NS	25 ± 4	30.5 ± 1.7	0.039
T stage [n (%)]				0.034			NS
Tis	3 (5.6)	1 (3.3)	2 (8.3)		0 (0)	0 (0)	
T1a	7 (13)	3 (10)	4 (16.7)		3 (11.5)	3 (15.8)	
T1b	5 (9.3)	2 (6.7)	3 (12.5)		2 (7.7)	3 (15.8)	
T2	6 (11.1)	2 (6.7)	4 (16.7)		2 (7.7)	3 (15.8)	
T3	17 (31.5)	10 (33.3)	7 (29.2)		7 (26.9)	6 (31.6)	
T4a	14 (25.9)	10 (33.3)	4 (16.7)		10 (38.5)	4 (21.1)	
T4b	2 (3.7)	2 (6.7)	0 (0)		2 (7.7)	0 (0)	
N stage [*n* (%)]				NS			NS
N0	26 (48.1)	13 (43.3)	13 (54.2)		11 (42.3)	9 (47.4)	
N1	7 (13)	2 (6.7)	5 (20.8)		1 (3.8)	4 (21.1)	
N2	11 (20.4)	7 (23.3)	4 (16.7)		7 (26.9)	4 (21.1)	
N3a	7 (13)	5 (16.7)	2 (8.3)		4 (15.4)	2 (10.5)	
N3b	3 (5.6)	3 (10)	0 (0)		3 (11.5)	0 (0)	
M stage [*n* (%)]				NS			NS
M0	52 (96.3)	28 (93.3)	24 (100)		24 (92.3)	19 (100)	
M1	2 (3.7)	2 (6.7)	0 (0)		2 (7.7)	0 (0)	
Number of LN [*n* ± SD]	12 ± 6	27 ± 13	25 ± 11	NS	29 ± 12.5	25 ± 12	NS
Number of positive LN (*n* (IQR)	0 (0)	0 (8)	1 (1.5)	NS	2.5 (8)	1 (0)	NS
Grade [*n* (%)]				NS			NS
1	7 (12)	3 (10)	4 (16.6)		2 (7.6)	2 (10.5)	
2	15 (27.8)	9 (30)	6 (25)		7 (26.9)	5 (26.3)	
3	32 (59.2)	18 (60)	14 (58.3)		17 (65.4)	12 (63.2)	
Perineural invasion [*n* (%)]				0.019			0.036
Yes	23 (42.6)	17 (56.7)	6 (25)		15 (57.7)	5 (26.3)	
No	31 (55.4)	13 (43.3)	18 (75)		11 (42.3)	14 (73.7)	
Lymphovascular invasion [*n* (%)]				NS			NS
Yes	30 (55.6)	20 (66.7)	10 (41.7)		17 (65.4)	9 (47.4)	
No	24 (44.4)	10 (33.3)	14 (58.3)		9 (34.6)	10 (52.6)	
Lauren type [*n* (%)]				NS			NS
Intestinal	31 (57.4)	17 (56.7)	10 (41.7)		15 (57.6)	9 (47.4)	
Diffuse	9 (16.7)	8 (26.7)	5 (20.8)		6 (23.1)	3 (15.8)	
Mixed	14 (25.9)	5 (16.7)	9 (37.5)		5 (19.2)	7 (36.8)	
Ulceration [*n* (%)]				NS			NS
Yes	39 (72.2)	21 (70)	18 (75)		19 (73.1)	16 (84.2)	
No	15 (27.8)	9 (30)	6 (25)		7 (26.9)	3 (15.8)	

IQR: inter quartile range; BMI: body mass index; LN: lymph nodes; NS: not significant.

**Table 2 biomolecules-11-01601-t002:** Correlations between leptin and clinicopathological predictors: Correlation matrix.

Variable Pearson Correlation Significance	Leptin Day 1	Leptin Day 7
Sex	0.188	0.408
	0.170	0.004 *
Location	0.008	0.045
	0.952	0.765
Type of operation	−0.057	−0.048
	0.682	0.750
BMI	0.668	0.494
	<0.0001 *	0.002 *
T	0.028	−0.189
	0.844	0.203
N	0.099	−0.133
	0.482	0.373
M	0.484	0.34
	<0.0001 *	0.019 *
Total number of extracted LNs	−0.201	−0.153
	0.149	0.305
Number of positive LNs	0.214	−0.062
	0.125	0.680
CRP day 1	0.280	0.323
	0.042 *	0.027 *
CRP day 7	0.347	−0.015
	0.017 *	0.925
Preoperative Hb	0.021	0.119
	0.881	0.424
Hb day 4	0.107	0.143
	0.597	0.514
Hb day 7	0.134	0.085
	0.358	0.583
Preoperative albumin	−0.089	0.026
	0.565	0.876
Albumin day 4	−0.221	0.159
	0.210	0.418
Albumin day 7	−0.026	0.063
	0.863	0.702
Perioperative CT	−0.335	−0.394
	0.053	0.28
Tumour grade	0.000	−0.312
	0.999	0.033 *
Lymphovascular invasion	0.04	−0.203
	0.779	0.171
Perineural invasion	0.105	0.014
	0.456	0.924
Lauren	−0.013	−0.121
	0.925	0.419
Ulceration	−0.139	−0.235
	0.321	0.112
Distal resection margin	0.039	−0.058
	0.784	0.702

BMI: Body mass index; T: TNM T stage; N: TNM N stage; M: TNM M stage; LNs: lymph nodes; Hb: Haemoglobin; CT: Chemotherapy; *: significant difference.

## Data Availability

Not applicable.

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
