# Peer review of "The Contribution of the Hunger Hormone Leptin in the Aetiology of Postoperative Anorexia after Laparoscopic and Open Gastrectomy in Gastric Cancer Patients"

_biomolecules, 2021, doi:10.3390/biom11111601_

Round 1

Reviewer 1 Report

The aim of this study was to compare leptin levels in patients with gastric cancer operated by either open surgery or laparoscopy. Leptin was measured at days 1, 4 and 7 post-surgery. The study was performed in 54 patients, 22 of them were subjected to laparoscopy and 32 to open surgery, respectively. The results show that leptin level was significantly lower in laparoscopy group, but only in patients with CRP below 100 mg/l. Type of surgery remained the independent predictor of post-operative leptin in multiple regression analysis.

The topic and the results are of interest but there are several concerns to be addressed.

1) It should be stated how many patients in each group were in the CRP <100 and >100 mg/l subgroups, respectively.

2) Fig. 2A: does p=0.038 refer to day 7 leptin level? Was preoperative leptin significantly different between groups or not? It would be more convenient to compare percent decrease in leptin between both groups rather than absolute levels.

3) It is unclear which variables were analyzed regarding correlations. The whole "correlation matrix" (r2 values for all pairs of variables) should be presented.

4) It is suggested that lower leptin in LS group may be responsible for less anorexia. However, leptin decreased in the postoperative period so its involvement in postoperative anorexia is difficult to be suggested.

5) Statistical analysis: it is stated that either t-test or Mann-Whitney U test were used. Were data examined for normality before choosing the appropriate test?

6) Line 202: R=0.104 is not the "strong relation"; please correct accordingly.

Author Response

Dear Editors, dear Reviewers,

Sincerest thanks for kindly offering us the valuable chance to revise our paper for publication in the journal Biomolecules, and for the very constructive, thoughtful, and insightful comments and suggestions. We have carefully revised our work accordingly, which has hopefully improved significantly in quality. We most sincerely wish that it could now be accepted for publication. Please find in the following pages our point-by-point responses to all comments and points; In the Revised Manuscript, modified places are underlined.

Thank you very much again for your time and kind work on our paper.

We are looking forward very much to hearing positively from you.

Best regards and best wishes,

Sincerely yours,

Tomaz Jagric

Reviewer No.1:

The aim of this study was to compare leptin levels in patients with gastric cancer operated by either open surgery or laparoscopy. Leptin was measured at days 1, 4 and 7 post-surgery. The study was performed in 54 patients, 22 of them were subjected to laparoscopy and 32 to open surgery, respectively. The results show that leptin level was significantly lower in laparoscopy group, but only in patients with CRP below 100 mg/l. Type of surgery remained the independent predictor of post-operative leptin in multiple regression analysis.

The topic and the results are of interest but there are several concerns to be addressed.

Comment No. 1: It should be stated how many patients in each group were in the CRP <100 and >100 mg/l subgroups, respectively.

Answer No. 1: Thank you very much for comment. We fully agree with the reviewer. We included the number of patients in “Results/Clinicopathological comparison between LS and OS” and in Table 1.

Comment No. 2: Fig. 2A: does p=0.038 refer to day 7 leptin level? Was preoperative leptin significantly different between groups or not? It would be more convenient to compare percent decrease in leptin between both groups rather than absolute levels.

Answer No. 2: We agree with the reviewer that the results presented in Figure 2a may be misleading. We therefore corrected the caption in figures 1a and 2a and added the comment:” Serum levels of leptin on day seven were significantly lower in the LS group.”

Preoperative levels of leptin were comparable in both groups (Table 1).

We agree with the reviewer that the percent of the decrease in leptin levels are more convenient. Therefore we introduced the relative concentration of leptin on day seven (Paragraph:” Correlations between leptin and clinicopathological predictors”): To adjust for the confounding effect of BMI on baseline leptin values, we determined the relative serum concentration of leptin on day seven. It was defined as the proportion of leptin levels on day seven compared to day one ([leptin day 7/leptin day 1]*100). The rate of leptin on day seven was not correlated significantly to either BMI or gender.

Comment No. 3: It is unclear which variables were analyzed regarding correlations. The whole "correlation matrix" (r2 values for all pairs of variables) should be presented.

Answer No. 3: Thank you very much for your very constructive suggestion which we fully agree on. We included the correlation matrix in the paragraph “Correlations between leptin and clinicopathological predictors” as Table 2.

Comment No. 4: It is suggested that lower leptin in LS group may be responsible for less anorexia. However, leptin decreased in the postoperative period so its involvement in postoperative anorexia is difficult to be suggested.

Answer No. 4: Thank you very much for comment. We agree with the reviewer’s opinion that postoperative anorexia is a complex and multifactorial process. Numerous studies have determined the pivotal role of leptin in the aetiology of postoperative anorexia, since it acts centrally in the arcuate nucleus of the hypothalamus, and inhibits the release of neuropeptide Y that, in turn, stimulates the food intake, decreases thermogenesis and increases plasma insulin and cortisone levels. The postoperative cytokine surge always increases the release of leptin contributing to anorexia after surgery. Hence, interventions that reduce serum leptin levels would be of immense value, since they could promote faster recovery of appetite. As the reviewer has pointed out, leptin levels were decreased after surgery, however, in patients operated with laparoscopic approach, the levels of leptin were 50% compared to preoperative levels and 74% in OS group (i.e. leptin levels were much lower in the LS group). Our conclusion was in line with the findings of Tuzun et al. They similarly found that increased levels of leptin were related to postoperative anorexia in ulcerative colitis. We agree that the precise mechanism has to be fully elucidated in future studies, but the relation of leptin levels, laparoscopic operation and better nutritional status after laparoscopy is clearly evident in our results.

Comment No. 5: Statistical analysis: it is stated that either t-test or Mann-Whitney U test were used. Were data examined for normality before choosing the appropriate test?

Answer No. 5: We agree with the reviewer. We thank for the constructive suggestion. The statistical methods did not sufficiently explain how the data was examined. Therefore we added:” The distribution of data was determined with Kolmogorov-Smirnov’s and Shapiro-Wilk’s tests of normality.” in “Methods/Statistical analysis”.

Comment No. 6: Line 202: R=0.104 is not the "strong relation"; please correct accordingly.

Answer No. 6: Thank you very much for your comment which we fully agree on. Although statistically significant, the term strong relation was poorly chosen. We have corrected accordingly in “Results/ Correlations between leptin and clinicopathological predictors”: significant relation.

Thank you very much for your very constructive, thoughtful, and insightful evaluations, and for your positive appraisal of and great support for our work!

Reviewer 2 Report

This study aimed to determine whether laparoscopic gastric cancer surgery (LS) attenuates postoperative anorexia by reducing serum leptin levels compared to open surgery (OS). Serum levels of leptin as well as markers of inflammation and nutritional status were compared between patients who received LS and those who received OS. Additionally, data were analyzed in the subgroup of patients with low inflammatory response (serum CRP < 100 mg/l). Major findings are that (1) although postoperative serum leptin levels are not different between LS and OS groups, they were significantly lower in LS compared to OS group in patients with CRP levels below 100 mg/l, (2) there was a strong correlation between serum CRP levels on day 1 and serum leptin levels on day 7, (3) serum albumin levels were significantly higher in LS compared to OS group in patients with CRP levels below 100 mg/l, and (4) analysis of the linear regression model showed that LS was associated with lower relative serum leptin levels (levels of day 7 relative to preoperative levels). The author concluded that LS causes less inflammatory response, which causes reduced serum leptin levels, promoting a better postoperative nutritional status (possible attenuated postoperative anorexia).

I have questions and suggestions which can be clarified and considered.

Specific points

  1. Some data presented in table 1 seem to be incorrect. For example, serum leptin levels on day 7 are 65 and 52 pg/ml in OS and LS groups, respectively. How could average leptin levels of all patients be 32.9 pg/ml? Same problems are seen in “Distal resection margin”, “CRP on day 1”, “Hemoglobin on days 4 and 7”, and “Albumin on day 7”.

  1. P7, L179-181: Which one is correct, 70% or 74%?

  1. P9, L201-206: I cannot find the scatter plot in Figure 2.

  1. Contribution of “gender” is studied in the present study. Did this study examine the effect of “gender” or “sex”? Please clarify.

  1. The author suggests the possible association between LS-induced low leptin levels and attenuated postoperative anorexia by showing an increased albumin levels at discharge as a marker of better nutritionals status (possibly as a consequence of improved nutrient intake). It would be ideal to show food intake data or other possible markers that links to energy intake more directly. Is postoperative body weight data available? It may be interesting to see if there is a better weight recovery in patients who received LS and had CRP levels below 100 mg/l compared to those received OS. This may support the attenuated anorexia in these patients.

6. Results of subgroup analysis suggest that LS can promote beneficial effect of postoperative nutritional status (or possibly anorexia) only when systemic inflammatory response is maintained at a low level. If systemic inflammatory response is high, LS may not be able to reduce relative leptin levels after the surgery. This may be discussed. 

Author Response

Dear Editors, dear Reviewers,

Sincerest thanks for kindly offering us the valuable chance to revise our paper for publication in the journal Biomolecules, and for the very constructive, thoughtful, and insightful comments and suggestions. We have carefully revised our work accordingly, which has hopefully improved significantly in quality. We most sincerely wish that it could now be accepted for publication. Please find in the following pages our point-by-point responses to all comments and points; In the Revised Manuscript, modified places are underlined.

Thank you very much again for your time and kind work on our paper.

We are looking forward very much to hearing positively from you.

Best regards and best wishes,

Sincerely yours,

Tomaz Jagric

Reviewer No. 2

This study aimed to determine whether laparoscopic gastric cancer surgery (LS) attenuates postoperative anorexia by reducing serum leptin levels compared to open surgery (OS). Serum levels of leptin as well as markers of inflammation and nutritional status were compared between patients who received LS and those who received OS. Additionally, data were analyzed in the subgroup of patients with low inflammatory response (serum CRP < 100 mg/l). Major findings are that (1) although postoperative serum leptin levels are not different between LS and OS groups, they were significantly lower in LS compared to OS group in patients with CRP levels below 100 mg/l, (2) there was a strong correlation between serum CRP levels on day 1 and serum leptin levels on day 7, (3) serum albumin levels were significantly higher in LS compared to OS group in patients with CRP levels below 100 mg/l, and (4) analysis of the linear regression model showed that LS was associated with lower relative serum leptin levels (levels of day 7 relative to preoperative levels). The author concluded that LS causes less inflammatory response, which causes reduced serum leptin levels, promoting a better postoperative nutritional status (possible attenuated postoperative anorexia).

I have questions and suggestions which can be clarified and considered.

Specific points

Comment No. 1: Some data presented in table 1 seem to be incorrect. For example, serum leptin levels on day 7 are 65 and 52 pg/ml in OS and LS groups, respectively. How could average leptin levels of all patients be 32.9 pg/ml? Same problems are seen in “Distal resection margin”, “CRP on day 1”, “Hemoglobin on days 4 and 7”, and “Albumin on day 7”.

Answer No. 1: We are thankful for your comment. We have corrected the average values.

Comment No. 2: P7, L179-181: Which one is correct, 70% or 74%?

Answer No. 2: Thank you very much for comment. We have corrected the value to: 74%.

Comment No. 3: P9, L201-206: I cannot find the scatter plot in Figure 2.

Answer No. 3: Thank you very much for your comment. We apologize for not including the scatterplot in the document. The scatterplot was added as Figure 2 and 3 (scatterplot leptin7/CRP and Leptin7/BMI) in the paragraph “Results/ Correlations between leptin and clinicopathological predictors.

Comment No. 4: Contribution of “gender” is studied in the present study. Did this study examine the effect of “gender” or “sex”? Please clarify.

Answer No. 4: Thank you very much for your very constructive and thoughtful comments which we fully agree on. The term gender is misleading and has been corrected to sex throughout the paper.

Comment No. 5: The author suggests the possible association between LS-induced low leptin levels and attenuated postoperative anorexia by showing an increased albumin levels at discharge as a marker of better nutritionals status (possibly as a consequence of improved nutrient intake). It would be ideal to show food intake data or other possible markers that links to energy intake more directly. Is postoperative body weight data available? It may be interesting to see if there is a better weight recovery in patients who received LS and had CRP levels below 100 mg/l compared to those received OS. This may support the attenuated anorexia in these patients.

Answer No. 6: We agree with the reviewer. Indeed the proposed monitoring of other markers of nutrient intake that would link leptin to energy intake would be highly desirable. Unfortunately the authors did not design the study to serially measure the body weight. We find this an intriguing proposal and will add the reviewer suggestion to the discussion. “In addition, monitoring of other markers that would link leptin to nutrient intake were not included in the present study. Weight recovery after operation could better determine the relation betwee nutrient intake and serum leptin levels.” was added as limitation of the study to discussion.

Comment No. 6: Results of subgroup analysis suggest that LS can promote beneficial effect of postoperative nutritional status (or possibly anorexia) only when systemic inflammatory response is maintained at a low level. If systemic inflammatory response is high, LS may not be able to reduce relative leptin levels after the surgery. This may be discussed.

Anwser No. 6: We fully agree with the reviewer’s opinion that this topic needs further discussion. Therefore “On the other hand, the results also suggest that any beneficial effect of laparoscopy on postoperative nutritional status can only be maintained as long as there are no postoperative complications. In patients with complications and increased CRP levels, the advantages on the nutritional status of laparoscopy over open surgery are lost.” was added to discussion.

Thank you very much for your very constructive, thoughtful, and insightful evaluations, and for your positive appraisal of and great support for our work!

Reviewer 3 Report

GENERAL COMMENTS

The manuscript addresses a topic of scientific interest, which is within the journal’s scope addressing an aspect of clinical relevance. Some more clinical implications might be discussed to provide a broader view. The manuscript may benefit from considering the following aspects:

Discussion:

The authors have focused only on leptin as a hormone modulating hunger, but other hormones and adipokines in particular are also involved in this process. For instance, ghrelin, an orexigenic hormone, might be even more important than leptin and should be at least mentioned in the Discussion. This is particularly important in patients with gastric cancer undergoing surgery, since depending on the type of surgery performed the ghrelin levels may decrease acutely (if the fundus is removed like in a total gastrectomy) or not change if the fundus integrity is maintained. 

In line with the previous point, possible additional hormonal changes should be also mentioned. Differences in adiposity imply changes in adipokines, some of which are well known to exert effects on inflammation insulin resistance and metabolism. The BMI of the patients studied was in the overweight category, were an increase in visceral adiposity and insulin resistance have been reported and alterations in the secretory pattern of adipokines takes place. In fact, some evident differences might be observed as regards the leptin-adiponectin ratio and some acute-phase reactants like serum amyloid A (ref Gómez-Ambrosi J, Salvador J, Rotellar F, et al. Increased serum amyloid A concentrations in morbid obesity decrease after gastric bypass. Obes Surg. 2006 Mar;16(3):262-9. doi: 10.1381/096089206776116525. PMID: 16545156.). This should be also commented in the Discussion.

Since leptin is a hormone with pleiotropic effects, a further aspect to be contemplated is how the changes observed after surgery might impinge on other biological processes like the potential influence on lipolysis (Frühbeck G, Gómez-Ambrosi J, Salvador J. Leptin-induced lipolysis opposes the tonic inhibition of endogenous adenosine in white adipocytes. FASEB J. 2001 Feb;15(2):333-40. doi: 10.1096/fj.00-0249com. PMID: 11156949).

Author Response

Dear Editors, dear Reviewers,

Sincerest thanks for kindly offering us the valuable chance to revise our paper for publication in the journal Biomolecules, and for the very constructive, thoughtful, and insightful comments and suggestions. We have carefully revised our work accordingly, which has hopefully improved significantly in quality. We most sincerely wish that it could now be accepted for publication. Please find in the following pages our point-by-point responses to all comments and points; In the Revised Manuscript, modified places are underlined.

Thank you very much again for your time and kind work on our paper.

We are looking forward very much to hearing positively from you.

Best regards and best wishes,

Sincerely yours,

Tomaz Jagric

Reviewer No. 3

GENERAL COMMENTS

The manuscript addresses a topic of scientific interest, which is within the journal’s scope addressing an aspect of clinical relevance. Some more clinical implications might be discussed to provide a broader view. The manuscript may benefit from considering the following aspects:

Discussion:

Comment no. 1: The authors have focused only on leptin as a hormone modulating hunger, but other hormones and adipokines in particular are also involved in this process. For instance, ghrelin, an orexigenic hormone, might be even more important than leptin and should be at least mentioned in the Discussion. This is particularly important in patients with gastric cancer undergoing surgery, since depending on the type of surgery performed the ghrelin levels may decrease acutely (if the fundus is removed like in a total gastrectomy) or not change if the fundus integrity is maintained.

Answer no. 1: We thank the reviewer for this insightful comment which we fully agree on. During the study design serum levels of ghrelin were also considered to be monitored in the study, as ghrelin has a much more direct influence on appetite, while the inverse proportional relation of leptin is somewhat contra intuitive. However, most of the ghrelin is secreted in the fundus of the stomach. If ghrelin would be an important mediator for postoperative appetite none of the patients would develop hunger after total gastrectomy. This, however, is not the case we observe in clinical practise. Even so, we agree with the reviewer that ghrelin has to be covered in the discussion. Therefore the following was in cluded in the paragraph “Discussion”: “Ghrelin has been extensively studied and has been found to have a prominent role in hunger modulation after bariatric operations. However, its role after gastric cancer surgery is questionable. Ghrelin is mainly produced in the gastric fundus, which is resected during total gastrectomy. Wang et al. have demonstrated a sharp decrease in serum ghrelin concentrations in the early postoperative period after gastrectomy and a slow increase in the following 360 days after surgery [25]. Therefore the role of ghrelin in early postoperative hunger and anorexia control should be negligible, which is not to say that other mediators are involved.”

Comment No. 2: In line with the previous point, possible additional hormonal changes should be also mentioned. Differences in adiposity imply changes in adipokines, some of which are well known to exert effects on inflammation insulin resistance and metabolism. The BMI of the patients studied was in the overweight category, were an increase in visceral adiposity and insulin resistance have been reported and alterations in the secretory pattern of adipokines takes place. In fact, some evident differences might be observed as regards the leptin-adiponectin ratio and some acute-phase reactants like serum amyloid A (ref Gómez-Ambrosi J, Salvador J, Rotellar F, et al. Increased serum amyloid A concentrations in morbid obesity decrease after gastric bypass. Obes Surg. 2006 Mar;16(3):262-9. doi: 10.1381/096089206776116525. PMID: 16545156.). This should be also commented in the Discussion.

Anwser No. 2: We fully agree and thank for the constructive comment. The following was added to discussion: “In both groups serum levels of leptin fell in the postoperative period. This was expected and in line with previous observations [22]. Gómez-Ambrosi et al. described a relation between leptin and other adipokines, specifically SSA that stimulates leptin secretion. SSA is produced mainly in the omentum. Omentectomy was performed as a part of radical gastrectomy in our study, whereby the predominant site of SSA synthesis was removed. The consequent drop of SSA production could explain the postoperative drop in serum leptin levels.”

and

“As leptin is produced mainly in adipose tissue [5-7], patients with higher BMI and women who have more adipose tissue, produce higher baseline serum levels of leptin. Gómez-Ambrosi et al. reported that obesity was associated with a state of low-grade chronic inflammation, whereby increased secretion of different adipokines was observed [22]. These might be directly responsible for the increased circulating leptin levels in obese patients. BMI and sex carried a confounding bias in our analysis, which was confirmed by a significant correlation between serum leptin, BMI and sex in our study.”

Comment No. 3: Since leptin is a hormone with pleiotropic effects, a further aspect to be contemplated is how the changes observed after surgery might impinge on other biological processes like the potential influence on lipolysis (Frühbeck G, Gómez-Ambrosi J, Salvador J. Leptin-induced lipolysis opposes the tonic inhibition of endogenous adenosine in white adipocytes. FASEB J. 2001 Feb;15(2):333-40. doi: 10.1096/fj.00-0249com. PMID: 11156949).

Anwser No. 3: We fully agree with the reviewer’s constructive comment. The precise mode of action of leptin should certainly be clarified in future studies. Its array of different effects makes it an exciting field of study. The following was added to discussion: “The precise mechanism by which hypoleptinemia exerts its activity on the nutritional status is complex due to the pleiotrophic effect of leptin [23]. Hypoleptinemia might directly stimulate nutrient intake, on the other hand, leptin may promote prosesses at least partly independent of its anorexic effect. Many reports have confirmed cross-talk between leptin, insulin and protein pathways [23]. Shimabukuro has reported that leptin constrains insulin biosynthesis and secretion in pancreatic β-cells and lower insulin [24]. Insulin is known to stimulate muscle cell protein synthesis [24]. This might explain lower albumin levels in patients with higher postoperative leptin in the OS group.”

Thank you very much for your very constructive, thoughtful, and insightful evaluations, and for your positive appraisal of and great support for our work!

Round 2

Reviewer 1 Report

The manuscript has been revised according to the reviewers' comments.

Reviewer 2 Report

Authors are responsible to my comments and I have no further concern.